# Towards Interpretable Natural Language Understanding with Explanations as Latent Variables

**Wangchunshu Zhou**[1*]   **Jinyi Hu**[2*]   **Hanlin Zhang**[3*]
**Xiaodan Liang**[4]   **Maosong Sun**[2]   **Chenyan Xiong**[5]   **Jian Tang**[6,7]
[1] Beihang University [2] Tsinghua University [3] South China University of Technology
[4] Sun Yat-sen University [5] Microsoft Research
[6] Mila-Québec AI Institute [7] HEC Montréal
zhouwangchunshu@buaa.edu.cn
hujy17@mails.tsinghua.edu.cn sms@tsinghua.edu.cn
{hlzhang109, xdliang328}@gmail.com
chenyan.xiong@microsoft.com
jian.tang@hec.ca

## Abstract

Recently generating natural language explanations has shown very promising results in not only offering interpretable explanations but also providing additional information and supervision for prediction. However, existing approaches usually require a large set of human annotated explanations for training while collecting a large set of explanations is not only time consuming but also expensive. In this paper, we develop a general framework for interpretable natural language understanding that requires only a small set of human annotated explanations for training. Our framework treats natural language explanations as latent variables that model the underlying reasoning process of a neural model. We develop a variational EM framework for optimization where an explanation generation module and an explanation-augmented prediction module are alternatively optimized and mutually enhance each other. Moreover, we further propose an explanation-based self-training method under this framework for semi-supervised learning. It alternates between assigning pseudo-labels to unlabeled data and generating new explanations to iteratively improve each other. Experiments on two natural language understanding tasks demonstrate that our framework can not only make effective predictions in both supervised and semi-supervised settings, but also generate good natural language explanations [2].

## 1 Introduction

Building interpretable systems for natural language understanding is critical in various domains such as healthcare and finance. One promising direction is generating natural language explanations for prediction [1–4], which has been shown very promising recently as they can not only offer interpretable explanations for back-box prediction systems but also provide additional information and supervision for prediction [5–7]. For example, given a sentence "*The only thing more wonderful than the food is the service.*", a human annotator may write an explanation like "*Positive, because the word 'wonderful' occurs within three words before the term food*", which is much more informative than the label "*positive*" as it explains how the decision was made. Moreover, the explanation can

---

[*] Equal contribution, with order determined by rolling a dice. Work was done during internship at Mila.
[2] Code is available at https://github.com/JamesHujy/ELV.git

serve as an implicit logic rule that can be generalized to other instances like "*The food is wonderful, I really enjoyed it.*"

There are some recent works [3, 4] that study generating natural language explanations for predictions and/or leverage generated explanations as additional features for prediction. For example, Camburu et al. [3] trained a language model to general natural language explanations for the task of natural language inference by training on a corpus with annotated human explanations. Rajani et al. [4] proposed a two-stage framework for common sense reasoning which first trained a natural language explanation model and then further trained a prediction model with the generated explanations as additional information. These approaches achieve promising performance in terms of both prediction performance and explainability. However, a large number of labeled examples with human explanations are required, which is expensive and sometimes impossible to obtain. Therefore, we are looking for an approach that makes effective prediction, offers good explainability, but requires a limited number of human explanations for training.

In this paper, we propose such an approach. We start from the intuition that the explanation-augmented prediction model is able to provide informative feedback for generating meaningful natural language explanations. Therefore, different from existing work which trains the explanation generation model and the explanation-augmented prediction model in separate stages, we propose to **jointly** train the two models. Specifically, taking the task of text classification as an example, we propose a principled probabilistic framework for text classification, where natural language **E**xplanations are treated as **L**atent **V**ariables (ELV). Variational EM [8] is used for the optimization, and only a set of human explanations are required for guiding the explanation generation process. In the E-step, the explanation generation model is trained to approximate the ground truth explanations (for instances with annotated explanations) or guided by the explanation-augmentation module through posterior inference (for instances without annotated explanations); in the M-step, the explanation-augmented prediction model is trained with high-quality explanations sampled from the explanation generation model. The two modules mutually enhance each other. As human explanations can serve as implicit logic rules, they can be used for labeling unlabeled data. Therefore, we further extend our ELV framework to an **E**xplantion-based **S**elf-**T**raining (ELV-EST) model for leveraging a large number of unlabeled data in the semi-supervised setting.

To summarize, in this paper we make the following contributions:

- We propose a principled probabilistic framework called ELV for text classification, in which natural language explanation is treated as a latent variable. It jointly trains an explanation generator and an explanation-augmented prediction model. Only a few annotated natural language explanations are required to guide the natural language generation process.
- We further extend ELV for semi-supervised learning (the ELV-EST model), which leverages natural language explanations as implicit logic rules to label unlabeled data.
- We conduct extensive experiments on two tasks: relation extraction and sentiment analysis. Experimental results prove the effectiveness of our proposed approach in terms of both prediction and explainability in both supervised and semi-supervised settings.

## 2   Related Work

Natural language (NL) explanations have been proved very useful for both model explanations and prediction in a variety of tasks recently [9, 10, 3, 4, 4]. Some early work [11, 9, 10] exploited NL explanation as additional features for prediction. For example, Srivastava et al. [9] converted NL explanations into classifier features to train text classification models. Fidler et al. [12] used natural language explanations to assist in supervising an image captioning model. Very recently, Murty et al. [10] proposed ExpBERT to directly incorporate NL explanations with BERT. However, most of these work require the explanations to be available in both training and testing instances, which is not realistic as annotating the explanation of a huge amount of instances is very time consuming and expensive. Moreover, the prediction becomes much easier once the explanations are given in the testing data.

There is some recent work that studied training a natural language explanation model and then used the generated explanations for prediction. For example, Camburu et al. [3] and Rajani et al. [4] proposed to separately train a model to generate NL explanations and a classification model that

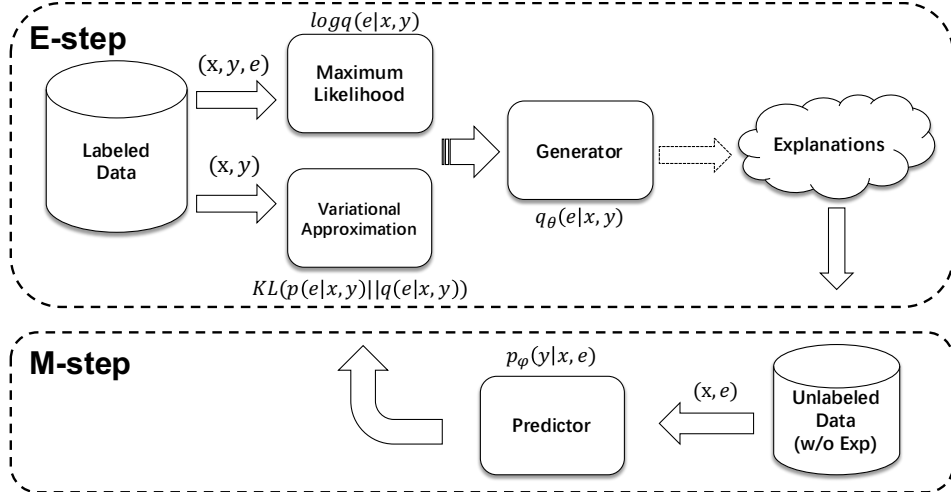

Figure 1: Overview of ELV. During E-step, we train our generator $p(e|x, y)$ to generate explanations given labeled data. For labeled data with annotated explanations (i.e. $\mathcal{D}_E$), we maximize the likelihood of the ground truth explanations. For labeled data without explanations (i.e. $\mathcal{D}_L$), we minimize the KL divergence between the variational distribution $q_\theta(e|x, y)$ and the ground truth posterior $p(e|x, y)$, which is calculated with the help of the prediction model. During M-step, we use the explanation generated in E-step to train the predictor $p(y|x, e)$ with MLE.

takes the generated explanations as additional input. Their approaches have shown very promising for improving the interpretability of classification models and increasing the prediction performance with explanations as additional features. However, their approaches require a large number of human annotated NL explanations to train the explanation generation model. Moreover, these approaches fail to model the interaction between generating NL explanations and exploiting NL explanations for prediction. As a result, there is no guarantee that the generated explanations reflect the decision-making process of the prediction model or beneficial to the prediction model. As reported by Camburu et al. [3] that interpretability comes at the cost of loss in performance. In this paper, we propose a principled probabilistic framework with explanations as latent variables to minimize the number of training instances with explanations by jointly training the natural language explanation module and the explanation-augmented prediction module.

Another relevant direction is treating natural language explanations as additional supervisions for semi-supervised learning instead of as additional features [13, 7]. For example, Hancock et al. [13] utilized a semantic parser to parse the NL explanations into logical forms (i.e., "labeling function"). The labeling functions are then employed to match the unlabeled examples either hardly [13] or softly [7] to generate pseudo-labeled datasets used for training models. However, these approaches require the explanations to be annotated in a form that can be accurately parsed by a semantic parser to form labeling functions, which may not be possible for many NLP applications. In our semi-supervised framework, semantic parsing is not required, and natural language explanations are interpreted with distributed representation obtained by pre-trained language models for labeling unlabeled data.

## 3 Methodology

### 3.1 Problem Definition

Given an input sentence $x$, we aim to predict its label $y$ and generate a natural language (NL) explanation $e$ describing why $x$ is classified as $y$. Specifically, given a few amount of training example with NL explanation annotation $\mathcal{D}_E = \{(x_1, y_1, e_1,), ..., (x_n, y_n, e_n,)\}$ and a relatively large set of labeled examples $\mathcal{D}_L = \{(x_{n+1}, y_{n+1}), ..., (x_m, y_m)\}$, our goal is to learn: 1) An explanation generation model $E_\theta$ that parametrize $q(e|x, y)$, which takes a labeled example $(x, y)$ as input and generates a corresponding natural language explanation $e$, and 2) an explanation-

augmented prediction model $M_\phi$, which parametrize $p(y|x, e)$ and takes an unlabeled example $x$ and NL explanations (as implicit rules) $E$ to assign a label $y$ to $x$.

## 3.2 Natural Language Explanation as Latent Variables

Given labeled data $(x, y)$, we treat the nature language explanation $e$ as a latent variable. For training, we aim to optimize the evidence lower bound (ELBO) of $\log p(y|x)$, which can be formulated as:

$$\log p(y|x) = \log \int p(e, y|x) de = \log \int q_\theta(e|x, y) \frac{p(e, y|x)}{q_\theta(e|x, y)} de \qquad (1)$$

$$\geq \mathcal{L}(\theta, \phi) = E_{q_\theta(e|x,y)} \log \frac{p(e, y|x)}{q_\theta(e|x, y)} = E_{q_\theta(e|x,y)} \log \frac{p_\phi(y|e, x) p(e|x)}{q_\theta(e|x, y)}, \qquad (2)$$

where $q_\theta(e|x, y)$ is the variational distribution of posterior distribution $p(e|x, y)$, $p(e|x)$ is the prior distribution of explanation $e$ for instance $x$, and $p_\phi(y|e, x)$ is the explanation-augmented prediction model.

Due to the large search space of natural language explanation $e$, instead of using reprametrization trick used in variational autoencoder [14], here we use the variational-EM algorithm for optimization. The overview of ELV is illuatrated in Figure 1. Note that in our training data, the explanations of a few number of labeled examples (i.e., $\mathcal{D}_E$) are given. Therefore, we first initialize the explanation generation model $E_\theta = q_\theta(e|x, y)$ and the prediction model $M_\phi = p_\phi(y|e, x)$ by training on $\mathcal{D}_E$ with maximum likelihood estimation (MLE). Then, we update the aforementioned models by maximizing the log-likelihood $\log p(y|x)$ using $\mathcal{D}_L \cup \mathcal{D}_E$ with variational-EM. In the variational E-step, we train the explanation generator to minimize the KL divergence between $q_\theta(e|x, y)$ and $p(e|x, y)$, which will be detailed in Section 3.3. In the M-step, we fix $\theta$ and $p(e|x)$ and update the parameters $\phi$ of the prediction model to maximize the log-likelihood $\log p(y|x)$.

## 3.3 E-step: Explanation Generation model

As the core component of our variational EM framework, the explanation generation model is expected to generate "soft" logic rules in the form of natural language explanations. However, training a seq2seq model to generate explanations of high quality from scratch is very challenging. Motivated by the recent finding that pretrained language models encode various types of factual knowledge and commonsense knowledge in their parameters [15–17], we employ UniLM [18]—a unified pre-trained language generation model that achieved state-of-the-art performance on many text generation tasks—as the explanation generation model $E_\theta$ in our framework. Specifically, the explanation generation model takes as input the concatenation of input sentence $x$ and the text description of its corresponding label $y$ to generate an explanation, which explains the label decision in natural language and can be treated as an implicit logic rule that can generalize to other examples.

Note that in the training data, only a small set of labeled examples are provided with explanations. Therefore, in the variational E-step, for labeled data $(x, y)$ without explanations (i.e. $\mathcal{D}_L$), we are trying to use the variational distribution $q_\theta(e|x, y)$ to approximate the ground truth posterior $p(e|x, y)$, which can be calculated as

$$p(e|x, y) \sim p_\phi(y|x, e) p(e|x) \qquad (3)$$

where $p_\phi(y|x, e)$ is parameterized by the prediction model and provides feedback for generating meaningful natural language explanations. We will introduce the detailed parametrization of $p(e|x)$ and $p_\phi(y|x, e)$ in the M-step.

For labeled data with explanations (i.e. $\mathcal{D}_E$), we just need to maximize the likelihood of the ground truth explanations. Therefore, the overall objective function of E-step can be summarized as:

$$O = \sum_{(x,y) \in \mathcal{D}_E} \log q(e|x, y) + \sum_{(x,y) \in \mathcal{D}_L} \mathrm{KL}(q(e|x, y) \| p(e|x, y)) \qquad (4)$$

## 3.4 M-step: Explanation-Augmented Prediction model

During M-step, the explanation-augmented prediction model is trained to predict the label of input sentence $x$ with the explanation $e$ generated from the variational distribution $q(e|x, y)$.

However, note that the label $y$ is not available during testing, and the explanations for the unlabeled $x$ can only be generated from the prior distribution $p(e|x)$. Therefore, there are some discrepancies between the distributions of the explanations for labeled data in the training stage and those for unlabeled data in the testing stage since generating a natural language explanation without conditioning on a label is harder. To mitigate this issue, in the prediction model, besides sampling an explanation from the variational distribution, we also add a set of explanations from $p(e|x)$, which retrieves a set of explanations from similar sentences.

Specifically, given an input sentence $x$ and the set of labeled and pseudo-labeled data consists of $(x', e', y')$, we retrieve $N$ explanations $\mathcal{E} := \{e'_i\}_{i=1}^N$ of which the corresponding sentences $x'$ are the most similar to the input sentence $x$, measured by the cosine similarity between the embedding of $x$ and each $x'$ from $\mathcal{D}_E$ under SentenceBERT [19], a pretrained sentence embedding model. Note that we do not directly use a seq2seq model to parametrize $p(e|x)$ because we find generating explanations without predicted labels often results in irrelevant and even misleading explanations.

---

**Algorithm 1:** Explanation-based Self-Training (ELV-EST)

---

**Input:** $\mathcal{D}_E = \{(x_1, y_1, e_1, ), ..., (x_n, y_n, e_n, )\}$, $\mathcal{D}_L = \{(x_{n+1}, y_{n+1}), ..., (x_m, y_m)\}$,
        unlabeled data $\mathcal{D}_U = \{x_{m+1}, ..., x_N\}$, Confidence threshold T
**Output:** $E_\theta(e|x, y), M_\phi(y|x, E)$
initialize $E_\theta$ and $M_\phi$ with $\mathcal{D}_E \cup \mathcal{D}_L$ using ELV
**repeat**
    |  **for** *each* $x_i \in \mathcal{D}_U$ **do**
    |     |  **if** $\max_y M_\phi(y|x, E) > T$ **then**
    |     |     | Assign pseudo-label $y_i$ to $x_i$ and generate explanation $e_i$ with $E_\theta$
    |     |     | Update $\mathcal{D}_L = \mathcal{D}_L \cup (x_i, y_i)$
    |     |     | Update $\mathcal{D}_E = \mathcal{D}_E \cup (x_i, y_i, e_i)$ (for explanation retrieval)
    |     |     | Update $\mathcal{D}_U = \mathcal{D}_U \setminus x_i$
    |     |  **end**
    |  **end**
    |  Train $E_\theta$ and $M_\phi$ on $\mathcal{D}_E \cup \mathcal{D}_L$ with ELV
**until** *Convergence or* $\mathcal{D}_U = \emptyset$

---

Let $\mathcal{E} = \{e_1, ..., e_n\}$ denotes all the explanations of $x$. For each $e_i \in \mathcal{E}$, we feed the explanation $e_i$ and the input sentence $x$, separated by a `[SEP]` token, to BERT [20] and use the vector at the `[CLS]` token to represent the interactions between $x$ and $e_i$ as a 768-dimensional feature vector:

$$\mathcal{I}(x, e_i) = \mathrm{BERT}([\texttt{[CLS]}; x; \texttt{[SEP]}; e_i]) \tag{5}$$

Our final classifier takes the concatenation of these vectors and outputs the final prediction as:

$$M_\phi(y|x, \mathcal{E}) = \mathrm{MLP}\left[\mathrm{Average}(\mathcal{I}(x, e_1); \mathcal{I}(x, e_2); ...; \mathcal{I}(x, e_n))\right] \tag{6}$$

At test time, for each unlabeled $x$, we first use $p(e|x)$ to retrieve a set of explanations and then predict a label with the explanation-augmented prediction model. Afterward, we can further employ the explanation generation model to generate an NL explanation to explain the prediction decision based on both the input sentence and the predicted label.

To summarize, by alternating between E-step and M-step where $q_\theta(e|x, y)$ and $p_\phi(y|e, x)$ are optimized respectively, the explanation generation model $E_\theta$ and the explanation-augmented prediction model $M_\phi$ are jointly optimized and mutually enhanced. Next, we describe how our framework can be applied to the semi-supervised setting where both human-annotated explanations and ground-truth labels are limited.

### 3.5 Explanation-based Self-Training

As natural language explanations can serve as implicit logic rules, which can generalize to new data and help assign pseudo-labels to unlabeled data. Therefore, we extend the ELV to the semi-supervised learning setting and propose an **E**xplanation-based **S**elf-**T**raining (ELV-EST) algorithm. In this setting, we only have limited labeled examples but abundant unlabeled data $\mathcal{D}_U = \{x_{m+1}, ..., x_N\}$.

Table 1: **Statistics of datasets.** We present the size of train/dev/test sets for 4 datasets in both supervised and semi-supervised settings. Moreover, # Exp means the size of initial explanation sets.

| Dataset | # Explanations | # Train (Supervised) | # Train (Semi-supervised) | # Dev | # Test |
|---|---|---|---|---|---|
| SemEval [21] | 203 | 7,016 | 1,210 | 800 | 2,715 |
| TACRED [22] | 139 | 68,006 | 2,751 | 22,531 | 15,509 |
| Laptop | 70 | 1,806 | 135 | 462 | 638 |
| Restaurant | 75 | 2,830 | 107 | 720 | 1,120 |

As illustrated in Algorithm 1, we first use ELV to initialize $E_\theta$ and $M_\phi$ with the limited labeled corpus $\mathcal{D}_E \cup \mathcal{D}_L$. Afterward, we iteratively use $M_\phi$ to assign pseudo-labels to unlabeled examples in $\mathcal{D}_U$ to extend the labeled data $\mathcal{D}_L$. We then use ELV to jointly train $E_\theta$ and $M_\phi$ with the augmented labeled dataset. At the same time, we also employ $E_\theta$ to generate new explanations with unlabeled examples and their pseudo-labels. In this way, we can harvest massive pseudo-labels and pseudo-explanations with unlabeled examples. The pseudo-labeled examples can be used to improve the models while also enable us to generate more NL explanations. In return, the newly generated explanations can not only improve the explanation generation model but also serve as implicit rules that help the prediction model assign more accurate pseudo-labels in the next iteration.

The proposed ELV-EST approach is different from the conventional self-training method in two perspectives. First, in addition to predicting pseudo-labels for unlabeled data, our method also discovers implicit logic rules in the form of natural language explanations, which in return helps the prediction model to better assign noisy labels to the unlabeled data. Second, our approach can produce explainable predictions with $E_\theta$. Compared to recent works [7, 13] that parse explanations to logic forms, our approach does not require task-specific semantic parsers and matching models, making it task-agnostic and applicable to various natural language understanding tasks with minimal additional efforts.

# 4 Experiments

## 4.1 Datasets

We conduct experiments on two tasks: relation extraction (RE) and aspect-based sentiment classification (ASC). For relation extraction we choose two datasets, TACRED [23] and SemEval [21] in our experiments. We use two customer review datasets, Restaurant and Laptop, which are part of SemEval 2014 Task 4 [24] for the aspect-based sentiment classification task. We use the human-annotated explanations collected in [7] for training our explanation-based models.

## 4.2 Experimental Settings

We conduct experiments in both the **supervised setting** where we have access to all labeled examples in the dataset and the **semi-supervised setting** where we only use a small fraction of labeled examples and considering the rest labeled examples in the original dataset as unlabeled examples by ignoring their labels. In both settings, only a few human-annotated NL explanations are available. The number of explanations, labeled data used in supervised/unsupervised setting, and the statistics of the datasets are presented in Table 1.

We employ BERT-base and UniLM-base as the backbone of our prediction model and explanation generation model, respectively. We select batch size over $\{32, 64\}$ and learning rate over $\{1e\text{-}5, 2e\text{-}5, 3e\text{-}5\}$. The number of retrieved explanations is set to 10 for all tasks. We train the prediction model for 3 epochs and the generation model for 5 epochs in each EM iteration. We use Adam optimizers and early stopping with the best validation F1-score.

## 4.3 Compared Methods

In the **supervised setting**, we compare ELV with the BERT-base baseline that directly fine-tunes the pre-trained BERT-base model on the target datasets. To show the importance of modeling the interactions between the explanation generation model and the explanation-augmented prediction

Table 2: Results (Micro-F1) on Relation Extraction datasets in supervised setting.

| Method | TACRED | SemEval |
|---|---|---|
| BERT$_{EM}$[25] | 66.3 | 76.9 |
| BERT$_{EM+MTB}$[25] | **67.1** | 77.5 |
| BERT-large | 66.4 | 78.8 |
| BERT-base | 64.7 | 78.3 |
| ELV (M-step only) | 65.4 | 80.2 |
| ELV (ours) | 65.9 | **80.7** |

Table 3: Results (Macro-F1) on ASC datasets in supervised setting.

| Method | Restaurant | Laptop |
|---|---|---|
| ASGCN [26] | 72.2 | 71.1 |
| BERT-PT [27] | 77.0 | 75.1 |
| BERT-SPC [28] | 77.0 | 75.0 |
| BERT-base | 75.4 | 72.4 |
| ELV (M-step only) | 76.2 | 74.1 |
| ELV (ours) | **77.8** | **75.2** |

Table 4: Results (Micro-F1) on Relation Extraction datasets in semi-supervised setting.

| Method | TACRED | SemEval |
|---|---|---|
| BERT-base | 25.1 | 49.3 |
| Pseudo-Labeling [29] | 28.6 | 50.2 |
| Self-Training [30] | 36.9 | 59.5 |
| Data Programming [13] | 25.8 | 47.9 |
| ELV-EST (ours) | **42.5** | **66.4** |

Table 5: Results (Macro-F1) on ASC datasets in semi-supervised setting.

| Method | Restaurant | Laptop |
|---|---|---|
| BERT-base | 32.2 | 34.6 |
| Pseudo-Labeling [29] | 42.5 | 38.2 |
| Self-Training [30] | 47.2 | 42.3 |
| Data Programming [13] | 38.2 | 36.3 |
| ELV-EST (ours) | **59.5** | **63.6** |

model, we also compare with a variant of our model, which only trains the explanation-augmented prediction module with all the explanations generated from the prior distribution, denoted as ELV (M-step only)). We also compare with some state-of-the-art algorithms on the RE and SA tasks.

In the **semi-supervised setting**, we compare ELV-EST against several competitive semi-supervised text classification methods including Pseudo-Labeling [29], Self-Training [30], and Data Programming [13] which incorporates NL explanations to perform semi-supervised text classification. Note that all compared model variants incorporate BERT-base as the backbone model.

### 4.4 Experimental Results

**Results on supervised setting.** We first present the results in the supervised setting in Table 2 and 3. ELV significantly outperforms the strong BERT baseline in all four datasets, demonstrating the effectiveness of exploiting NL explanations as additional information for natural language understanding. ELV also consistently outperforms the ELV (M-step only), showing that ELV's variational EM training effectively models the interactions between explanation and prediction. Also, the performance of ELV compares favorably against several competitive recent studies focusing on RE and ASC respectively, further demonstrating the effectiveness of ELV.

**Results on semi-supervised setting.** The results in the semi-supervised setting are presented in Table 4 and 5. In the semi-supervised scenario, ELV-EST method significantly outperforms various semi-supervised text classification methods, as well as the data programming approach. The latter uses pre-defined rules to parse the NL explanations into logic forms and match unlabeled examples, on all four datasets. The improvement upon the BERT-base + self-training baseline is around 7 points for RE datasets and over 15 points for ASC datasets in terms of F1 score. This demonstrates the effectiveness of ELV-EST in the semi-supervised setting.

**Results on explanation generation.** We further evaluate the quality of the explanation generation model with human evaluation. We invite 5 graduate students with enough English proficiency to score the explanations generated on the test set with input sentences and the labels predicted by the explanation-augmented prediction module[3]. The annotation scenarios include the explanantions' informativeness (Info.), correctness (Corr.), and consistency (Cons.) with respect to the model prediction. The inner-rater agreement is at 0.51 Kappa score. The details of human evaluation and examples of generated explanations are presented in the Appendix due to space constraints.

Table 6: Human evaluation results. The scores scale from 1 to 5 (the larger, the better). The inner-rater agreement measured by Kappa score is 0.51.

| Model | Inf. | Corr. | Cons. |
|---|---|---|---|
| Seq2Seq | 2.43 | 3.27 | 2.68 |
| Transformer | 2.35 | 3.12 | 2.62 |
| UniLM | 3.48 | 3.94 | 3.14 |
| ELV (ours) | **3.87** | **4.20** | **3.51** |

Table 7: Results on ASC datasets with explanations with words randomly corrupted (80%). Orig + Rand Exp is the 1:1 mix of original and randomly corrupted explanations.

| Method | Restaurant | Laptop |
|---|---|---|
| BERT-base | 75.4 | 72.4 |
| w. 80% Rand Word | 73.2 | 70.9 |
| Orig + Rand Exp | 76.9 | 74.0 |
| ELV (ours) | **77.8** | **75.2** |

For comparison, we include a fine-tuned UniLM model with annotated NL explanations, as well as two baselines trained from scratch using annotated NL explanations, one with a vanilla transformer model and the other an attention-based LSTM seq2seq model. The results are in Table 6. The explanations generated by our ELV framework are substantially better than those generated by the fine-tuned UniLM model. ELV generates better NL explanations that are relevant to the model's decision-making process, because it models the interactions of the explanation generation model and the prediction model.

## 4.5 Analysis

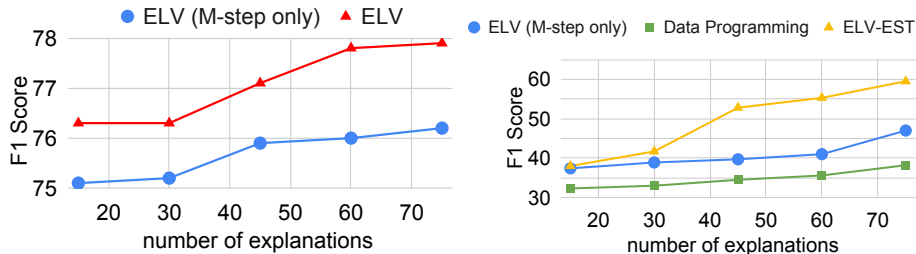

Figure 2: Performance with different number of explanations. We compare our method with baseline(s) in both supervised setting (left) and semi-supervised setting (right).

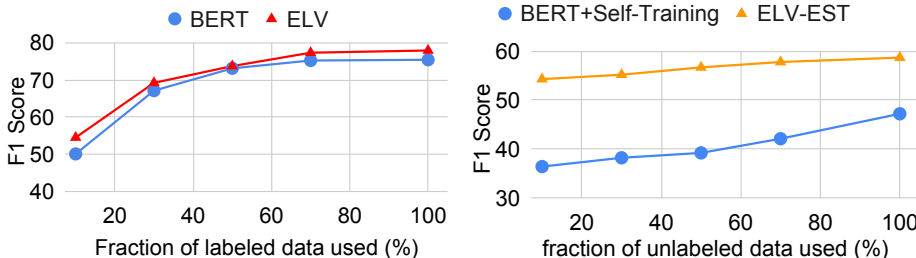

Figure 3: Performance with different number of labeled or unlabeled data in supervised setting (left) and semi-supervised setting (right) respectively.

**Performance with corrupted explanations.** We first investigate the model performance w.r.t. the quality of the retrieved explanations. We compare with corrupted explanations which randomly replace 80% of the words in the original explanations, results shown in Table 7. The performance with corrupted explanations significantly decreases as expected. The high-quality explanations help the model better generalize while the random ones may confuse the model.

**Performance with different numbers of explanations.** We then investigate the performances with different amounts of explanations. As illustrated in Figure 2 (left), with as few as 15 annotated explanations, ELV significantly outperforms its counterpart trained without the variational EM framework in the supervised setting. The performance of ELV continues to improve with more

explanations but the performance of ELV (M-step only) starts to saturate with 45 explanations, showing the importance of modeling the interactions between the explanation generation model and explanation-augmented prediction model. Similar results are observed in the semi-supervised learning setting in Figure 2 (right).

**Performance with different numbers of labeled/unlabeled data.** We also investigate the performance of different models with different proportions of training data. From Figure 3 (left), we can see that ELV consistently outperforms the BERT baseline with different amounts of labeled data. Especially, the improvement is the most significant when only 10% of labeled data is used. This is because human explanations provide additional supervision and can serve as implicit logic rules to help generalization. In the semi-supervised learning setting (Figure 3, right), ELV-EST outperforms traditional self-training methods by a large margin, especially with fewer unlabeled data, further confirming the improved generalization ability from explanations.

## 5   Conclusion

In this paper, we propose ELV, a novel framework for training interpretable natural language understanding models with limited human annotated explanations. Our framework treats natural language explanations as latent variables that model the underlying reasoning process to enable interactions between explanation generation and explanation-based classification. Experimental results in both supervised and semi-supervised settings show that ELV can not only make effective predictions but also generate meaningful explanations. In the future, we plan to apply our framework to other natural language understanding tasks. In addition, we plan to test the effectiveness of our framework on other pre-trained language models that are either stronger (e.g., XLNet [31], RoBERTa [32], ALBERT [33], ELECTRA [34], etc.) or more computationally efficient. (e.g., DistilBERT [35], BERT-of-Theseus [36], DeeBERT [37], PABEE [38], etc.)

## Broader Impact

Deep learning has achieved great success in natural language understanding. However, most existing systems are not interpretable, which limit their applications to many domains such as healthcare, finance, and legislation. In these domains, interpretability is a high priority. This paper proposed a principled probabilistic model for text classification, which not only makes effective prediction but also offers good explainability. Though the model is developed for the task of text classification, it is a very general framework and could be generalized to other tasks in natural language understanding. Such a system could be useful in a variety of tasks such as decision making with clinical notes in healthcare, justice, and criminal identification with legal data, and risk management in finance.

On the other hand, such a system also brings potential risks depending on the quality of the generated natural language explanations. For example, the generated natural language could have certain biases, which have been reported in many natural language understanding systems [39, 40]. How to mitigate these risks will be our future work. Another potential risk is that the explanation generation model in our framework generates ad-hoc explanations that are not necessarily informative about how the model makes its predictions, since the model can come up with whatever explanation it thinks would pair with its predicted label. This is a common drawback for current explanation generation models. Our framework partially mitigates this problem since the generated explanations are in return used in the training process of the explanation-augmented classifier through the explanation retrieval process.

## Acknowledgments

This project is supported by the Natural Sciences and Engineering Research Council (NSERC) Discovery Grant, the Canada CIFAR AI Chair Program, collaboration grants between Microsoft Research and Mila, Samsung, Amazon Faculty Research Award, Tencent AI Lab Rhino-Bird Gift Fund and a NRC Collaborative R&D Project (AI4D-CORE-06). The authors would like to thank Meng Qu, Ziqi Wang, Cheng Lu, and the anonymous reviewers for their valuable feedback and insightful comments.

## Footnotes

[3]The prediction module is jointly trained with the explanation generation module

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
