[Supplementary Material · Appendix.pdf]

## A  Examples of Explanations

In this section we present several human annotated natural language explanations and explanationos generated by the explanation generation model in the ELV framework in different datasets to better understand the proposed approach.

### A.1  TACRED

#### A.1.1  Human Annotated Explanations

a) *Although not a Playboy Playmate , she has appeared in nude pictorials with her Girls Next Door costars and fellow Hefner girlfriends Holly Madison and Kendra Baskett(OBJ), then known as Kendra Wilkinson(SUBJ).*

   **Label:** `per:alternate_names`

   **Explanation:** The term "then known as" occurs between SUBJ and OBJ and there are no more than six words between SUBJ and OBJ

b) *Burke(SUBJ) 's mother Melissa Bell(OBJ) was a singer in the dance group Soul II Soul, which had hits in the 1980s and 1990s.*

   **Label:** `per:parents`

   **Explanation:** SUBJ and OBJ sandwich the phrase "'s mother" and there are no more than three words between SUBJ and OBJ

c) *Ellen Pompeo(OBJ) secretly married Chris Ivery(SUBJ) Congratulations to the newlyweds and let them live happily ever after !!*

   **Label:** `per:spouse`

   **Explanation:** There are no more than four words between SUBJ and OBJ and SUBJ and OBJ sandwich the phrase "secretly married"

#### A.1.2  Machine Generated Explanations

a) *Lomax shares a story about Almena Lomax, his mother and a newspaper owner and journalist(OBJ) in Los Angeles , taking her(SUBJ) family on the bus to Tuskegee, Ala., in 1961.*

   **Label:** `per:title`

   **Explanation:** the word "family" is right after SUBJ.

b) *What happened to their investments was of no interest to them, because they would already be paid , said Paul Hodgson(OBJ), senior research associate at the Corporate Library(SUBJ), a shareholder activist group.*

   **Label:** `per:org:top_members/employees`

   **Explanation:** The word "senior research associate at" appears right before SUBJ.

c) *Iroquois passport dispute raises sovereignty issue The National Congress of American Indians(SUBJ), based in Washington, DC(OBJ), has advocated on behalf of the lacrosse team, urging British officials to allow the members entry into England on their Iroquois-issued passports.*

   **Label:** `org:country_of_headquarterse`

   **Explanation:** The word "based in" appears right before OBJ.

### A.2  SemEval

#### A.2.1  Human Annotated Explanations

a) *Morton's SUBJ-O is the most common cause of localized OBJ-O in the third interspace and these diagnostic tests produce good indications of the condition.*

   **Label:** `Cause-Effect(e1,e2)`

   **Explanation:** Between SUBJ and OBJ the term "is the most common cause of" appears and SUBJ precedes OBJ

b) *The frontal SUBJ-O is a part of the OBJ-O that maintains very close ties with the limbic system.*
   **Label:** `Component-Whole(e1,e2)`
   **Explanation:** Between SUBJ and OBJ the term "is a part of the" appears and SUBJ precedes OBJ.

c) *Out current Secretary is gathering SUBJ-O from past OBJ-O and committee chairs.*
   **Label:** `Entity-Origin(e1,e2)`
   **Explanation:** The phrase "from past" links SUBJ and OBJ and there are no more than three words between SUBJ and OBJ and OBJ follows SUBJ.

## A.2.2 Machine Generated Explanations

a) *SUBJ-O caused OBJ-O at the Charlotte Douglas International Airport Monday morning.*
   **Label:** `Cause-Effect(e1,e2)`
   **Explanation:** There is only one word "caused" between subj and obj and obj follows subj.

b) *The base in which it sits hides the damage which occurred when the SUBJ-O was removed from its initial OBJ-O on the Long Tan battlefield.*
   **Label:** `Entity-Origin(e1,e2)`
   **Explanation:** Between subj and obj the phrase "was removed into" occurs and there are no more than four words between SUBJ and OBJ and SUBJ precedes OBJ

c) *A hinge SUBJ-O attaches a OBJ-O pivotally to a base of an electronic device and has a pivoting leaf and a stationary leaf.*
   **Label:** `Component-Whole(e1,e2)`
   **Explanation:** The phrase "attaches a" between SUBJ and OBJ and OBJ follows SUBJ

## A.3 Restaurant

## A.3.1 Human Annotated Explanations

a) *We had great desserts (including the best cannoli I've ever had) and then they offered an after dinner drink, on the house. (Term: cannoli)*
   **Label:** `positive`
   **Explanation:** The word "best" directly precedes the term.

b) *All the desserts the group tried got favorable reviews. (Term: desserts)*
   **Label:** `positive`
   **Explanation:** The string "favorable" appears no more than 5 words after the term.

c) *The most annoying thing, though, is the fact that the servers seem to be trained to drive revenue. (Term: servers)*
   **Label:** `negative`
   **Explanation:** The word "annoying" occurs before the term.

## A.3.2 Machine Generated Explanations

a) *This little place is wonderfully warm welcoming. (Term: place)*
   **Label:** `positive`
   **Explanation:** The terem is followed by "wonderful".

b) *The falafal was rather over cooked and dried but the chicken was fine. (Term: chicken)*
   **Label:** `positive`
   **Explanation:** The word "fine" occurs within 3 words after the term.

c) *Service was awful - mostly because staff were overwhelmed on a Saturday night. (Term: staff)*
   **Label:** `negative`
   **Explanation:** The word "unbearable" occurs within three words after the term.

Table 8: Results (Macro-F1) on ASC datasets in supervised setting with different fraction of labeled data used.

| Fraction of labeled data used | 60% | 70% | 80% | 90% | 100% |
|---|---|---|---|---|---|
| ELV (ours) | 75.5 | - | - | - | - |
| BERT-base | 74.6 | 75.0 | 75.3 | 75.4 | 75.4 |

## A.4 Laptop

### A.4.1 Human Annotated Explanations

a) *The DVD drive randomly pops open when it is in my backpack as well, which is annoying. (Term: DVD drive)*

**Label:** `negative`

**Explanation:** The string "annoying" occurs after the term

b) *The Apple team also assists you very nicely when choosing which computer is right for you. (Term: Apple team)*

**Label:** `positive`

**Explanation:** The string "very nicely" occurs after the term by no more than 6 words.

c) *The design is awesome, quality is unprecedented. (Term: design)*

**Label:** `positive`

**Explanation:** The word "awesome" is within 2 words after the term.

### A.4.2 Machine Generated Explanations

a) *I ordered my 2012 mac mini after being disappointed with spec of the new 27 Imacs. (Term: spec)*

**Label:** `negative`

**Explanation:** The word "disappointed" occurs within 3 words before the term.

b) *I found the mini to be exceptionally easy to set up. (Term: set up)*

**Label:** `positive`

**Explanation:** The phrase "exceptionally easy" occurs within 3 words before the term.

c) *However, there are MAJOR issues with the touchpad which render the device nearly useless. (Term: tourchpad)*

**Label:** `negative`

**Explanation:** The phrase "nearly useless" occurs within 3 words after the term.

## B Human Evaluation Details

For human evaluation, we random sample 100 examples in the test set of the Restaurant dataset and use ELV to predict the labels of the selected examples. Then we use different compared models to generate explanations of the prediction results with both the input sentences and the predicted labels. Afterward, we invite 5 graduate students with English proficiency to score the explanations. The annotation scenarios include the explanantions' informativeness(Info.), correctness (Corr.), and consistency (Cons.) with respect to the model prediction. Specifically, the informativeness measures to what extent the generated explanation is helpful to understand the model's prediction output. The correctness measures whether the explanation is factually correct (e.g. word "good" before the terms leads to positive label while word "annoying" is negative). The consistency refers to whether the explanation is consistent with the input sentence (i.e. the description in the explanation is true w.r.t the input sentence).

## C  Addition Analysis Experiments

In this section, we report additional experimental results comparing the performance of ELV with 60% of labeled data and the performance with the BERT-base model with 60%, 70%, 80%, 90%, and 100% of labeled data to investigate to what extent human annotated explanations can replace human labeled examples. The result is shown in Table 8. We find that ELV can achieve or even exceed the performance of a BERT-base model trained with much more labeled data. This confirms that ELV can effectively leverage human annotated explanations as additional information.