[Reviews · NeurIPS 2020]

Review 1

Summary and Contributions: This paper proposes an EM framework for usage of natural language explanations to improve downstream classification tasks, specifically relation extraction and sentiment analysis. The advantages of this framework are in not requiring large amounts of generated explanations. They are also able to demonstrate effective operations in semi-supervised settings, where labeled data may be sparse. The method is mathematically sound and the results are promising.

Strengths: The method is a novel contribution in using EM algorithm for explanation generation, and seems highly relevant to the NeurIPS community. 1. The method is theoretically well grounded, using expectation maximization to effectively generate explanations without requiring large amounts of gold standard human generated explanation data. The framework also has the ability to extend to the semi-supervised setting, demonstrating improved performance over baseline counterparts in the same setting. 2. The approach performs reasonably well at the prediction task has explanations and doesn’t require too much labeled human explanation data. Compared to some prior work, the work does not require semantic parsing and utilize pre-trained language models for labeling unlabeled data. 3. The framework is fairly general and can be applied to various classification related NLU tasks.

Weaknesses: 1. The method is theoretically sound, however, there is no clear proof that the quality of generated explanations is high. The cohen’s kappa of human checked experiments is 0.51, which is considered very weak. Stronger quantitative results would be needed to show that the explanations are of high quality, and some qualitative results could solidify that result. 2. Some tables could use completeness in terms of baseline comparisons. For example, I wonder how [29],[30],[13] fit into table 2. Even though the methods are specifically for semi-supervised settings, the supervised table could be treated as one end of the spectrum of labeled data available. 3. The split used for semi-supervised setting in Table 1 seems rather arbitrary. Were there experiments that guided those chosen splits? 4. There is no mention of state of the art in either tasks. It would be great to see how this model compares to state of the art in all tasks. If the framework does not perform as well, perhaps there could be commentary about semi-supervised performance of those models. The choice of baseline comparisons seems rather arbitrary. 5. While the experiments in Fig 2 and Fig 3 demonstrate the level of semi-supervised data needed to perform well, the comparisons don’t seem complete. Which dataset and task are each tables representing? Why are only a few baseline models compared in these tables?

Correctness: Yes, for the most part. The methodology is empirically sound, utilizing standard EM algorithm and variational lower bound to optimize their objective.

Clarity: The paper is really clearly written, and provides specific explanations of their contributions. It is very easy to read, and navigate.

Relation to Prior Work: Yes the paper describes the relationship to the prior work very clearly, and the contributions of the paper are very apparent from those comparisons.

Reproducibility: No

Additional Feedback: 1. Some tables could use more explanations. What is the difference between the two middle rows in Table 7? There only seems to be a short description on line 255, which does not talk about orig + rand exp. 2. A possible future work could be: Is it possible to augment existing task data (full training and test splits) with some noisy unlabeled data extracted from other datasets/ MTurk to demonstrate the true effectiveness of semi-supervised setting without artificially and randomly choosing which part of the full dataset should be chopped of from training? 3. The corrupted explanations seem to produce results not too different from weaker baselines (that aren’t operating based on explanations). This makes me wonder about whether the quality of explanations generated by this model are that high. 4. What is BERT+Self Training in Figure 3 (right)? This would need more explanation. 5. Edit to the future work: assessing the quality of the generated explanations still seems to need a lot of work. The Kappa is low, showing that the quality of human annotations is also not great.


Review 2

Summary and Contributions: The authors propose learning explanations for subsequent downstream NLP tasks in a semi-supervised learning approach. The approach addresses the challenge of obtaining annotated data of interpretable explanations. Contributions: the authors propose using the variational EM framework for joint optimization for both the explanation generation model and the explanation-augmented prediction model. The unlabled data is annotated with pseudo-labels generated from the generation model in a self-supervised/ self-training framework.

Strengths: The proposed approach tackles a realistic setting in which human annotation is not always available. In particular, human annotation of natural language explanations can be expensive to obtain. 

The adaptation of a variational EM approach for joint learning both explanation generation model and explanation-augmented prediction model is innovative. This self-supervised approach learns pseudo-labels of unlabeled data.

Weaknesses: The authors might want to articulate the technical contribution in more detail. How does the proposed EM approach differ from others in addition to generating natural language explanations? The number of samples with explanations is very limited compared with samples without explanations (e.g. 139 vs 106,046 in TACRED datasets). Should one be concerned that the inferred explanations are being learning from an under-sampled pool of data? Or is this not necessarily a concern for the tasks at hand? More detailed analysis would be helpful for readers to understand the effectiveness of the approach better.

Correctness: The authors claimed that " The two modules mutually enhance each other" in the Introduction but it was not immediately clear how they do so. It is described how explanation generation model can support explanation-augmented prediction model through the generation of pseudo-labels. However, it is not clearly detailed how the enhancement in the reverse direction happens. It would be useful to describe more in the Introduction how this process is done in the proposed approach (an example is the last sentence in section 3.4)

Clarity: The paper is well written. However, some justification may be required to interpret natural language explanations with distributed representations as this might affect the model interpretability and explanability. Some prior work has used semantic parsing to parse explanations into logical forms. This work offers to first represent explanation as latent variables to support prediction models, which is then used to fully generate a natural language explanation. The author could describe more about how the model interpretability in the new approach is different from prior work. Will the generated natural language explanation in this case be counted as an inference induced from the predicted label itself rather than intepreted as an explanation of a prediction decision?

Relation to Prior Work: Explanation-based related work is clearly discussed in the paper. It would be more informative if authors can discuss in more detail relevant work that applies EM in similar settings, e.g. semi-supervised learning with limited labels.

Reproducibility: Yes

Additional Feedback: The authors could describe in more detail the generation of natural language explanation after obtaining the predicted label. Did you use the same language model backbone used during the E-step? Using beam search or greedy decoding? Did you use the latent variable e generated from the variational distribution?


Review 3

Summary and Contributions: This paper proposes an EM approach to semi-supervised explanation generation. In this setup, a model has access to a large set of training examples for a text classification task, of which a small subset of these are annotated with ground-truth explanations. The idea is to train both an explanation generator and predictor in this setting. The proposed model ELV seems to work well, particularly at generating explanations.

Strengths: At least to the best of this reviewer's knowledge, this idea seems novel and elegant. The results on the considered datasets seem strong as well. The analysis in Section 4.5 is also useful, particularly the dependence on the amount of labeled versus unlabeled data, and the helpfulness of both steps of the EM approach.

Weaknesses: * The main issue, to this reviewer, is that the datasets chosen seem somewhat simplistic in their explanations. This shows in some of the example explanations (in the supplemental) like "The word "best" directly precedes the term." -- which function on the word level, and thus are probably pretty simple for a model to learn (and use as an internal code when forcing the label to be recoverable by the predictor). To this reviewer, it would be more exciting to compare with a dataset such as e-SNLI [3] or Cos-E [4], since those seem to require more complex reasoning. * Slight: it would be helpful to see the dependence on the number of retrieved explanations (set to 10 everywhere). === Update: thanks for the response! I read through the other reviewers' responses and the rebuttal. Adding in e-SNLI (or some more interesting datasets) would greatly improve the paper to this reviewer, and I also advise following the suggestions raised by R4. I still recommend acceptance though.

Correctness: Yes (modulo some concerns about whether these datasets are interesting).

Clarity: Yes, at least to this reviewer.

Relation to Prior Work: Yes, at least to this reviewer.

Reproducibility: Yes

Additional Feedback:


Review 4

Summary and Contributions: The authors propose a method to improve the performance of natural-language-explanation based classifiers by treating the NL explanation as a latent variable. Furthermore, instead of only optimizing end-to-end, the latent explanations are explicitly optimized to match ground-truth explanations where they exist. This ostensibly helps ground the explanation in some way (as opposed to the full e2e approach, in which the two components of the model could collude and produce uninterpretable or arbitrary latent representations). The paper shows how to use this method in a semi-supervised learning setup to yield performance gains in situations where few ground-truth explanations are available, and subjectively evaluate the quality of the explanations.

Strengths: The method of leveraging explanations for semi-supervised learning seems clever, and the results for the case with few ground-truth explanations are quite good.

Weaknesses: My main concern is about how explanations are being employed as latent variables. I had assumed based on the introduction that the final predictor would factor through the final explanation. This would provide the faithfulness guarantee that two inputs which produce the same explanation would produce the same output label. However, it seems that during training, the explanation is conditioned on the gold label. The paper points out on L161 that “generating explanations without a predicted label often results in irrelevant and even misleading explanations.” But if this is the case, then why would we expect explanations to be useful or faithful as latent variables? Then in L167–170, the paper clarifies that the final explanation for the prediction is also conditioned on the label. This now sounds exactly like post-hoc explanation generation—we didn’t actually gain any explainability over a model which didn’t have the whole latent variable rigamarole. So what was it all about? It seems to me that the retrieval of explanations of nearby inputs is the important bit. This paper strikes me as not being about explanation at all, but about improving a classifier’s learning signal (especially in the semi-supervised case) by retrieving nearest neighbors and featurizing them in a clever way (which happens to look on its surface like “generating explanations”). Indeed, the strongest results are improvements on supervised and semi-supervised performance. With an inter-annotator agreement score of 0.51 and the general ambiguity over what makes an explanation useful, I’m not sure what to take away from the explanation evaluation results. So this work seems worthwhile to me, but it’s not clear that it’s making a contribution to explainable ML so much as leveraging “explanations” to improve ML. Given all of that, I can’t begrudge this paper too much on the issue for a reject because there are worse offenders in the published literature. But I would highly recommend reframing and rewriting the paper to account for this problem. EDIT: Thank you for your comments in the response. However, I feel the core issue of my review was not addressed. You mention that generating explanations ad-hoc (or post-hoc, I guess) "enables generating more informative explanations" (L40), but the crucial question is: informative about what? These explanations, even if more aesthetically pleasing or accurate to what humans wrote on average, are in all likelihood *less* informative about why the model made the classification decision that it did, which is the point of interpretable or explainable ML. I can see how it makes sense to use gold labels in training to approximate the posterior for the explanation generator—that is clever and nice. But it seems to me that conditioning the final explanation on the gold label totally gives up the interpretability argument. It would be much more solid if the latent explanation were used instead. If that is apparently lower quality, fine—it just means the model isn't as good at explaining its decisions as it is at fooling us with nice-sounding post-hoc rationalizations. This issue is not just nitpicking. It's crucially important for the real-world usefulness of explainable ML, and messing it up leads directly to the worst-case scenario of ML systems which fool their users into thinking they're doing something reasonable when their decisions are actually wrong, biased, and harmful. Consider your example on L25, and suppose a model labeled it correctly with the explanation you provide. Now consider the hypothetical from my original review: “if anything here is wonderful, its food sure isn’t.” A smart model may then produce the correct label with a more detailed explanation (as you suggest on L50 of your response). But this would make the original explanation *misleading*—it would lead me to wrongly predict the model's behavior. And if the model is simply free to come up with whatever explanation it thinks would pair with its predicted label, there is no force moving the model towards providing explanations that are not misleading. It's entirely post-hoc and has nothing to do with interpretability. This is why I very strongly recommend either walking back the claims in the paper about interpretability, or changing the final model and evaluation to only use the latent explanations. Even these are still not great, because the predictor sees both x and e, we don't have any guarantees about explanatory power (much better if it didn't see x at all); this caveat would still need to be mentioned. But at least there would be some kind of loss that pushes the model in the right direction. Again, I know that most papers on explanation generation fail the standard I'm setting out, which is why I'm not recommending this paper for rejection. (I don't think this implies the standard is too strong; there is no strong-enough standard of scholarly integrity, especially when it comes to the central claims of a line of research.) But please, please seriously consider being more careful with claims about interpretability. Otherwise the standard will never improve.

Correctness: As far as I can tell, yes.

Clarity: Yes. A couple of gripes though: I think the bottom-right element in Figure 1 is mislabeled. It says “Labeled data (w/o Exp)” but it seems to be input/explanation pairs, without labels. Furthermore, I think this paper needs to reckon more carefully with what it means for an explanation to be good or useful. For example, consider the explanation on L25, and the hypothetical review: “if anything here is wonderful, its food sure isn’t.” The given explanation sounds more like a surface pattern than human reasoning, and can be fooled in the same way. Generated explanations are often referred to as “implicit logic rules,” but when they are generated entirely post-hoc, the connection to logical rules is very dubious. I think this constitutes a pretty big overclaim and should be avoided.

Relation to Prior Work: Yes.

Reproducibility: Yes

Additional Feedback: Regarding broader impact: I feel it is addressed reasonably, at least under the view of their work as explainable ML. However, I am concerned that the work does not actually increase the explainability of why the model made the decisions it did (rather, it just is better at producing reasonable-looking explanations as post-hoc justifications), while it pitches itself as doing so. This strikes me as dangerous, because it could promote the use of what are still black-box or biased systems with a false veneer of accountability.

[Author Response · NeurIPS 2020]

We would like to thank all the reviewers for your insightful and constructive reviews and your commendation for the theoretical soundness, novelty, and effectiveness of our method. Here are the responses to some of your concerns:

**Reviewer #1:** (1) **Concerns on the quality of generated explanations.** Good point! The cohen's kappa score of 0.51 is moderate. Kappa scores for human evaluation of natural language generation tasks are generally moderate (between 0.4-0.6) as it is relatively difficult for human annotators to accurately rate machine-generated text. We also conducted another quantitative human evaluation based on pairwise comparison and observe similar results. We have presented some qualitative results (generated explanations) in the Appendix. (2) **Merge the results of supervised and semi-supervised setting.** Thanks for the suggestion. We will merge this in the revised version. (3) **Data Splits, SoTA, and Baselines.** We follow the dataset split used in [7] throughout all experiments. We chose the baselines that use the same backbone model and did not include the ones with additional resources such as knowledge bases. Our model compares similarly or slightly worse compared with the SoTA results that leverage additional resources or larger backbone models. We will add the SoTA results for reference in the revised version. (4) **Clarification for Fig2&3 .** The results in Fig2&3 are conducted on the Restaurant datasets of the ASC task and are compared against the most competitive baselines. (5) **More clarification on the experiments**. orig + rand exp refers to the model with the combination of original explanations and the corrupted explanations. The corrupted explanations (w. 80% Rand Word) is worse than the baseline model, showing the importance of good explanations. The BERT+self-training baseline denotes the baseline BERT-base model trained with the self-training method.

**Reviewer #2:** (1) **Technical Contributions.** We develop a novel EM algorithm to jointly train a natural language explanation generation module and explanation-augmented prediction module, which mutually enhance each other. Specifically, the explanation-based classifier can help improve the explanation generator as it is able to identify some high quality of explanations for training the generator (see Eq.(4)). (2) **Small samples of explanations.** The number of annotated explanations is indeed small because we want to focus on the more realistic scenario where only a few annotated explanations are available. However, the annotated explanations cover all the relation types/sentiments in RE/ASC tasks so that annotated explanations are still representative. (3) **Details on the explanation generation.** For generating NL explanation after prediction, we use the variational distribution $q(e|x, y)$ for generating the explanation given the input sentence $x$ and predicted label $y$. The same language model backbone and beam search are used for the generation.

**Reviewer #3:** (1) **Experiments on Data Sets e-SNLI [3] or Cos-E [4]**. Thanks very much for pointing out these very relevant data sets. We will add the results on these data sets in the revised version. (2) **Performance w.r.t. Number of Retrieved Explanations.** This is a good point. We change the number of retrieved explanations to 5 and 15, and test them on the Restaurant dataset. The F1 are 77.3 and 75.1 respectively, which are worse than 77.8 with 10 explanations reported in this paper. We will add the sensitivity analysis w.r.t. the number of retrieved explanations in the revised version.

**Reviewer #4:** (1) **Explanations as Latent Variables**. Note that our goal lies in both generating explanations and providing extra supervision through natural language explanations. Treating explanations as latent variables allow the learning and inference process to be connected in a principled way via explanation generation and explanation-augmented prediction. The label-based explanation used during training is actually used for approximating the posterior distribution of explanations for training the explanation generator. The predictor is based on retrieved explanations that are not based on the label of the input. The final explanation of a prediction is ad-hoc (generated based on both the input and the prediction), this enables generating more informative explanations. The explanation retrieval process is important for the predictor in M-step since it conveys additional supervision. Therefore, explanations not only serve the role of explaining but also serve as additional supervision. For the human evaluation of the quality of generated explanations, a kappa coefficient of 0.51 is ok for evaluating natural language generation results as it is hard for human annotators to perfectly rate machine-generated texts. Thanks for the caveat, we admit the necessity of investigating other aspects of explainability to ensure our model's alignment with societal desiderata and leave debiasing, risks mitigation and adversarial defense as our future work. (2) **Overclaiming on explanations**: In our generic explainable framework, we use existing explanations in [7] for simplicity. Following works like [7,13], we use "logic rules" and "labeling function" interchangeably since they convey information about labels. Our model could potentially learn to explain during testing in the example you provided if faithful natural language explanations in more complicated logical forms are provided, e.g. sent: "anything here is wonderful but the food isn't", exp: The word "wonderful" and "but" occur before the term "food" and "isn't" occurs after the term "food". We'll clarify this issue in the final version to avoid overclaiming. Thanks for the comments and we have revised the label in Fig 1 accordingly in our next draft.

[Meta-Review · NeurIPS 2020]

This paper proposes an EM framework for explainable language processing. Strength • The idea is new and neat. • The proposed method is technically sound. • Experiments are conducted to support the claims • The paper is generally well-written. Weakness • The experiment part and presentation can be further improved. The authors are suggested to further improve the quality of the paper based on the reviewers' comments. NOTE FROM PROGRAM CHAIRS: For the camera-ready version, please expand your broader impact statement to include a more substantive discussion on the potential negative impacts of your work, as well as mitigations. One reviewer has noted that "This work will raise ethical concerns if the generated explanations are very incorrect, especially if deployed in healthcare. There is no solid quantitative result/ metric for the quality of the explanations produced in the work."